# The relationship between advanced glycation end products and gestational diabetes: A systematic review and meta-analysis

**Mekonnen Sisay** [1]*, **Dumessa Edessa** [2], **Tilahun Ali** [3], **Abraham Nigussie Mekuria** [1], **Alemu Gebrie** [4]

**1** Department of Pharmacology and Toxicology, School of Pharmacy, College of Health and Medical Sciences, Haramaya University, Harar, Ethiopia, **2** Department of Clinical Pharmacy, School of Pharmacy, College of Health and Medical Sciences, Haramaya University, Harar, Ethiopia, **3** Department of Psychiatry, School of Nursing and Midwifery, College of Health and Medical Sciences, Haramaya University, Harar, Ethiopia, **4** Department of Biomedical Sciences, School of Medicine, Debre Markos University, Debre Markos, Ethiopia

* mekonnensisay27@yahoo.com

**Data Availability Statement:** Additional files are attached as supplements along with the manuscript.

## Abstract

### Introduction

Gestational Diabetes Mellitus (GDM) is a condition in which women without history of diabetes experience hyperglycemia during pregnancy, especially at the second and third trimesters. In women who have had GDM, an elevated body mass index (BMI) may have a substantial impact for persistent hyperglycemia in their lives after gestation. Beyond hyperglycemia, increased local oxidative stress directly promotes the formation of Advanced Glycation End-products (AGEs). Hence, this systematic review and meta-analysis was aimed to determine the relationship between the level of AGEs and/or related metabolic biomarkers with GDM.

### Methods

Literature search was carried out through visiting electronic databases, indexing services, and directories including PubMed/MEDLINE (Ovid®), EMBASE (Ovid®), google scholar and WorldCat to retrieve studies without time limit. Following screening and eligibility evaluation, relevant data were extracted from included studies and analyzed using Rev-Man 5.3 and STATA 15.0. Inverse variance method with random effects pooling model was used for the analysis of outcome measures at 95% confidence interval. Hedge's adjusted g statistics was applied to calculate the standardized mean difference (SMD) to consider the small sample bias. Besides, meta-regression, meta-influence, and publication bias analyses were conducted. The protocol has been registered on PROSPERO with ID: CRD42020173867.

### Results

A total of 16 original studies were included for the systematic review and meta-analysis. Compared with women with pregnant controls, the level of AGE was significantly higher in women with GDM (SMD [95% CI] = 2.26 [1.50–3.02], Z = 5.83, P < 0.00001; $I^2$ = 97%, P<

**Funding:** The authors received no specific funding for this work.

**Competing interests:** The authors have declared that no competing interests exist.

**Abbreviations:** AGE, Advanced glycation End Products; BMI, Body Mass Index; GDM, Gestational Diabetes Mellitus; HOMA-IR, Homeostatic Assessment Model for Insulin Resistance; ICAM-1, Intercellular Adhesive Molecule 1; MeSH, Medical Subject Headings; ROS, Reactive Oxygen Species; SMD, Standardized Mean Difference; VCAM-1, Vascular Cell Adhesive Molecule 1.

0.0001). The BMI was also significantly higher in women with GDM (SMD [95% CI] = 0.97 [0.33–1.62], Z = 2.98, P = 0.003) compared to controls. Regarding specific and related metabolic biomarkers, there was higher level of HOMA-IR (SMD [95% CI] = 0.39 [0.22–0.55], Z = 4.65, P < 0.0001, after sensitivity analysis) and HbA1c (SMD [95% CI] = 0.58 [0.03–1.12], Z = 2.07, P = 0.04, after sensitivity analysis) in gestational diabetic women. Subgroup analyses indicated that studies conducted in Asia and Europe, at third trimester of pregnancy and blood/plasma AGE samples showed a significant difference in AGE level among women with GDM compared to pregnant controls. What is more, meta-regression with the sample size (regression coefficient (Q) = -0.0092, P = 0.207) and year of publication (Q = 0.0035, P = 0.984) suggested that the covariates had no significant effect on the heterogeneity.

## Conclusion

The study indicated that there was a strong relationship between AGE and GDM. Besides, the BMI and other specific biomarkers showed a significant difference between the two groups indicating the high risk of developing long-standing type 2 diabetes and its complications in gestational diabetic women. Early detection of these biomarkers may play a pivotal role in controlling postpartum diabetic complications.

## Introduction

According to the American Diabetes Association (ADA), gestational diabetes mellitus (GDM) can be defined as glucose intolerance with onset during pregnancy and typically resolves itself postpartum. It is treated as a major public health concern due to its adverse maternal and neonatal outcomes and a likelihood of developing type-2 diabetes later in the lives of the mothers and offsprings [1]. The genetic, epigenetic, and environmental factors may jointly contribute to the development of GDM. Hence, the underlying mechanisms involved in the pathogenesis of GDM remain complex and gradually evolving. Available evidence indicated that chronic inflammation, oxidative stress, gluconeogenesis, and placental factors contribute to the pathology of GDM [2]. Even if pregnancy is normally considered as a state of oxidative stress, the presence of GDM heightens the oxidative state. The rise in the levels of reactive oxygen species (ROS) has been associated with non-enzymatic glycation of macromolecules which may partly play a role in the development of postpartum type 2 diabetes mellitus and maternal and neonatal complications [3, 4].

Under favorable conditions, a series of non-enzymatic reactions occur between the amino groups of macromolecules and the carbonyl groups of reducing sugars, a process known as Maillard reaction. Such early glycation adducts undergo further rearrangement into final stable heterogeneous products called advanced glycation end products (AGEs) [5, 6]. Such reactions alter the structure and function of macromolecules leading to pathological aging processes. To this end, hyperglycemic and oxidative stress conditions accelerate this process [6]. AGEs can chemically be classified as fluorescent cross-linking AGEs (e.g. pentosidine and crossline), non-fluorescent cross-linking AGEs (e.g. arginine–lysine imidazole cross-links), and non-cross-linking AGEs (e.g. N-carboxymethyl-lysine (CML)) [7].

The formation of AGEs normally occurs both exogenously and endogenously. The exogenous production of AGEs occurs when foods are processed with high temperature. It is evident that fried food items such as cookies, biscuits, and chips having one or more AGE-forming

ingredients are overwhelmed with high levels of AGEs [5, 8–10]. The contemporary lifestyle provides a conducive environment for thermally processed food items replete with pro-inflammatory and oxidative stress-inducing AGEs. By stimulating appetite and causing overnutrition, such food items pose a risk for overweight and obesity [8, 11]. A study indicated that a dietary quality index score was negatively correlated with the serum levels of some AGEs [12] emphasizing the quality of processed food items determines their AGE content. Likewise, higher level of maternal consumption of fried fish and fried chicken just before conception was associated with an increased risk of GDM [13]. A case-control study conducted in Iran indicated that western dietary pattern was associated with an increased risk of GDM [14]. Therefore, wise dietary adjustment has a paramount importance for controlling an AGE load in the body. The endogenous formation of AGEs is also heightened in the presence of hyperglycemia and oxidative stress, the two hallmarks of diabetes creating a vicious cycle and hence the causal relationship remains "a chicken-egg dilemma". AGEs also contribute to the development of diabetes through augmenting further oxidative stress and AGE receptor (RAGE) mediated downstream signaling [15]. AGEs mediate inflammatory actions via protein kinases and the nuclear factor kappa B (NF-kB) signaling pathway in human gestational tissues [16]. The growing body of evidence has indicated AGEs-RAGE interaction elicits oxidative stress which in turn triggers proliferative, inflammatory, thrombotic, and fibrotic reactions. This evidence supports AGEs involvement in diabetes and age-related disorders [6, 7, 17]. In this regard, Dariya and Nagaraju summarized the role of AGE-RAGE interaction and downstream signaling pathways highly implicated for tumorigenesis and diabetic complications. The generation of ROS, activation of NF-kB, and protein kinases play a pivotal role for downstream signaling processes. Activation of NF-kB in turn upregulates the RAGE and perpetuates the signaling process [18]. The authors also pointed out phytochemical constituents such as genistein and curcumin can sequester highly reactive dicarbonyl compounds such as methylglyoxal and glyoxalase thereby prevent the formation of AGEs and their interaction with RAGE [18]. Piuri et al also elucidated the possible involvement of new inflammatory and metabolic biomarkers (Methylglyoxal, glycated albumin, PAF, and TNF-α) in the mechanisms related to GDM complications and exploration into the vicious cycle connecting inflammation, oxidative stress, and AGEs [19]. What is more, AGEs may lead to abnormal expressions of tight junction-associated integral membrane proteins (ZO-1 and Occludin) in vascular endothelial cells of placenta via RAGE/NF-kB signaling pathway, thereby abolishing the integrity of the membrane and increasing placental permeability [20]. In view of individual studies, they had low statistical power for inference and a sort of inconsistency in reporting findings about the relationship between the level of AGEs and/or related biomarkers with GDM. Hence, we conducted this systematic review and meta-analysis to generate pooled estimates at global level.

## Methods

### Study protocol and registration

The Preferred Reporting Items for Systematic Review and Meta-analysis (PRISMA) guideline was used for screening and eligibility assessment of identified studies for systematic review and meta-analysis [21]. This systematic review and meta-analysis was conducted by following the PRISMA Protocol [22]. Besides, the contents of this systematic review and meta-analysis have been well reported in the completed PRISMA checklist [23] (**S1 Table**). The study protocol has been registered on the International Prospective Register of Systematic Reviews (PROSPERO) with unique ID: CRD42020173867 and available at: https://www.crd.york.ac.uk/PROSPERO/display_record.php?ID=CRD42020173867&ID=CRD42020173867

## Data sources and search strategy

An electronic search was performed on legitimate databases, indexing services, and search engines including PubMed/ MEDLINE (Ovid), EMBASE (Ovid), Google scholar, and World-Cat with predefined keywords, indexing and MeSH terms until March 31st, 2020. By removing the non-explanatory terms from the research question, the keywords and MeSH terms were connected with Boolean operators in search of legitimate databases as follows without time limit (["advanced glycation end products" OR "AGEs" OR "advanced glycosylation end products" OR "glycosylation end products, advanced" [MeSH]] AND ["gestational diabetes" [MeSH] OR gestational* OR pregnancy OR "pregnancy induced diabetes" OR "diabetes in pregnancy"]). References of identified citations and Google Scholar were also searched to identify additional studies. Truncation was used when appropriate to fine-tune the search and increase the number of relevant findings.

## Inclusion and exclusion criteria

During the screening and eligibility assessments, there were predefined inclusion and exclusion criteria to include relevant studies. Observational studies (Case-control, cohort or cross-sectional) addressing the level of AGEs and/or related metabolic biomarkers among pregnant women with GDM (cases) and normally progressing pregnancy (controls) were included. Restriction was not applied on the years of publication and geographical location, but only studies written in English language were considered for inclusion. Review papers, editorials, commentaries, opinions, and case reports were excluded during screening of titles and abstracts. Studies addressing the AGEs in women with GDM without control and animal-based preclinical studies were excluded during the selection process. We also excluded cases mixed with other types of diabetes during eligibility assessment. Studies with irretrievable full texts (after requesting full texts from the corresponding authors via email and/or Research Gate accounts) or studies with unrelated or insufficient outcome measures or studies with ambiguous outcomes of interest were excluded.

## Screening and eligibility of studies

The reference lists identified from different sources were exported to ENDNOTE version 7.2 software (Thomson Reuters, Stamford, CT, USA) with compatible formats. Studies retrieved from various databases were combined. Duplicate records were removed with the help of ENDNOTE software followed by careful visual inspection by considering distinct referencing styles of sources which the software could not detect as duplicate. Each record was independently assessed by two authors (MS and TA) using the predefined inclusion and exclusion criteria stated above. Following initial screening of records with their titles and abstracts, rigorous assessment of full texts was made by MS and DE. Disagreement raised among authors at any phase of the work was solved by discussion with the rest authors.

## Data extraction

Important data were extracted from included studies using Excel sheet (S2 Table).The authors (MS and ANM) independently extracted the data related to study characteristics and outcome measures: including first author, publication year, study design and population, study setting and country, body mass index (BMI) (Mean ± standard deviation (SD)), stage of pregnancy, sample size (sum of cases and controls), type of samples collected, mean level of AGEs and specific metabolic biomarkers including homeostatic assessment for insulin resistance (HOMA-

IR), intercellular/vascular cell adhesion molecule -1 (ICAM-1/VCAM-1), and glycated hemoglobin (HbA1c) in both groups with specific units of measurement (Mean ± SD or Mean [95% confidence interval]).

## Critical appraisal of studies (risk of bias assessments)

Following the assessment of eligible articles, two authors (MS and AG) independently assessed the methodological validity and analysis of outcome measures using the Joanna Briggs Institute (JBI) critical appraisal checklist for observational studies, University of Adelaide, Australia [24]. The assessment tool consisted of design-specific questions about the quality of the study based on the following responses: Yes, No, Unclear, and Not Applicable. This critical appraisal was conducted to assess the internal and external validity of studies and to determine the extent to which each study has addressed the possibility of bias in its design, conduct, and analysis. The mean score of the two authors was taken for final decision and studies with a score of 'Yes' greater than or equal to half of the respective number of appraisal questions were included in the study.

## Outcome measurements

Our primary outcome of interest was the relationship between the level of overall AGEs with GDM. Subgroup analyses were performed based on trimester of pregnancy, geographic location, and type of samples. Additional meta-analyses were run to find out the association between BMI ($kg/m^2$) as well as clinically relevant metabolic biomarkers such as HOMA-IR, HbA1c, and ICAM-1/VCAM-1 with GDM.

## Data processing and analysis

The extracted data were exported from Excel to Rev-Man 5.3 software (Cochrane Organization, England) for analysis of overall outcome measures, subgroups, and publication bias. Meta-regression and meta-influence analyses were conducted using STATA 15.0 software (Stata Corporation, College Station, TX, USA). Considering the variation in true effect sizes across the population, inverse variance (IV) method with random effects pooling model was applied for meta-analysis at 95% confidence interval. Considering instrumental variation and small sample bias, the standardized mean difference (SMD) was calculated with Hedge's adjusted 'g' statistics [25]. The heterogeneity of studies was assessed using $I^2$ statistics. The "leave-one-out" sensitivity analysis was carried out to assess outliers that likely have a substantial impact on the overall effect size and between-study heterogeneity [26]. Influence analysis was conducted to evaluate whether a single study significantly affected the pooled SMD estimate [27]. The presence of publication bias was determined by using Egger's regression test and visualization of funnel plot asymmetry [28, 29]. The pooled estimate was declared statistically significant based on Z statistics and a cutoff point of $p < 0.05$ (two-sided).

## Results

### Search results

A total of 472 studies were retrieved through visiting legitimate databases, indexing services, search engines, and repositories. From these, 94 duplicate studies were identified and removed using ENDNOTE and careful visual inspection. Then, 378 records were retained for further screening using their titles and abstracts. Among which, a total of 341 records (234 studies by titles and 107 studies by abstracts) were excluded. The full texts of the remaining 37 studies

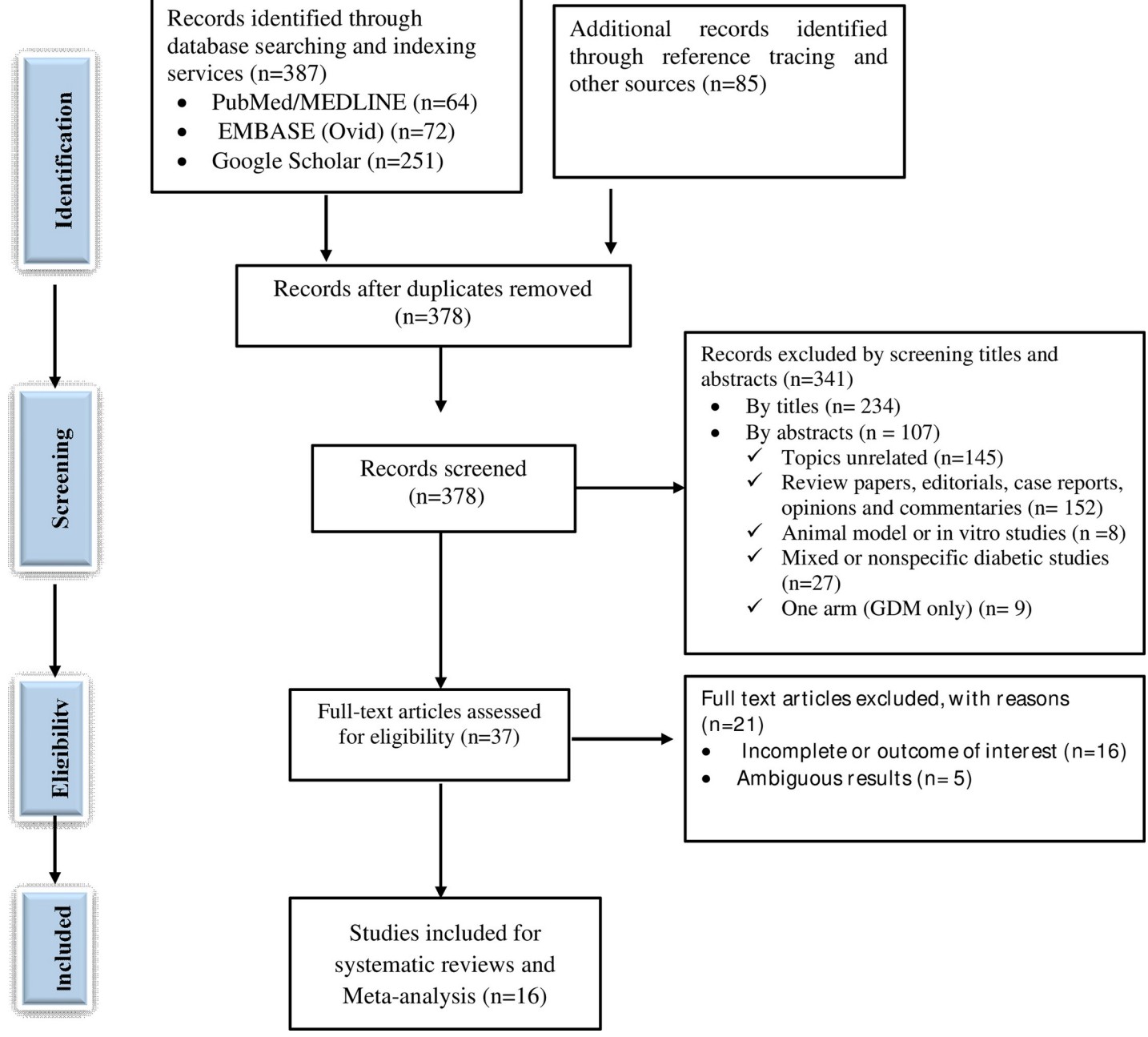

**Fig 1. PRISMA flow diagram showing the selection process of retrieved studies.**

were assessed for eligibility and 21 of which were excluded with various reasons. Finally, 16 studies were included for systematic review and meta-analysis (**Fig 1**).

## Results of critical appraisal

Rigorous appraisal of included observational studies (case-control and cross-sectional studies with 10- and 9-point scales, respectively) resulted in average quality scores ranging between 5 and 10. Fortunately, all included studies fulfilled the minimum criteria and retained for systematic review and meta-analysis (**S3 Table**).

## Study characteristics

A total of 16 studies (with one study retained for specific AGE measures and metabolic bio-markers) were included for systematic review and meta-analysis. The publication years of included studies ranged from 2008 to 2019. Regarding the geographical distribution, the review included seven studies from Europe [30–36], seven studies from Asia [37–43] and two studies from Latin America [44, 45]. Thirteen of these studies employed case-control design [30, 32, 33, 35–44] whereas the rest three studies were cross-sectional in design [31, 34, 45]. Specifically, the meta-analysis of AGEs involved 774 cases and 834 controls. The systematic review and meta-analysis involved studies that reported the outcome measures at both first and second trimesters [32], first and third trimesters [36], second trimester [30, 31, 34, 41, 44], second and third trimesters [38, 42], and third trimester of pregnancy [33, 35, 37, 39, 40] whereas two studies did not specify gestational period during sample collection [43, 45]. Blood sample was collected from 13 studies [30, 31, 33, 35–40, 42–45], skin sample from two studies [32, 34], and placenta sample from one study [41] (**Table 1**). For analysis of specific AGEs and metabolic biomarkers, five and four studies were included to generate the pooled estimates of HOMA-IR [31, 37, 42, 43, 45], and HbA1c [31, 33, 37, 43], respectively. Three studies with four effect measures (including one repeated measure at two gestational time points) were included for ICAM-1/VCAM-1 analysis [39, 40, 45] (**Table 2**).

## Meta-analysis of outcome measures

Fourteen effect measures with similar measurement units (AU/mL) (obtained from 11 studies involving three repeated measures) were included for meta-analysis of the overall AGE levels in the body. Compared with the control group, the level of AGE was significantly higher in women with GDM (SMD [95% CI] = 2.26 [1.50–3.02], Z = 5.83, P < 0.00001; Tau$^2$ = 1.95, I$^2$ = 97%, P< 0.0001) (**Fig 2**). The BMI (Kg/m$^2$) was also significantly higher in women with GDM (SMD [95% CI] = 0.97 [0.33–1.62], Z = 2.98, P = 0.003) compared to normally progressing pregnant controls (**Fig 3**). Regarding specific and related metabolic biomarkers, there was higher level of HOMA-IR (SMD [95% CI] = 0.63 [0.24–1.03], Z = 3.13, P = 0.002 and SMD [95% CI] = 0.39 [0.22–0.55], Z = 4.65, P < 0.0001 for before and after sensitivity analysis, respectively) (**Fig 4**) and HbA1c (SMD [95% CI] = 1.55 [0.36–2.75], Z = 2.55, P = 0.01 and SMD [95% CI] = 0.58 [0.03–1.12], Z = 2.07, P = 0.04 for before and after sensitivity analysis, respectively) in gestational diabetic women (**Fig 5**). Despite showing higher magnitude in GDM, the level of ICAM-1/VCAM-1 did not show significant difference between women with GDM and controls (SMD [95% CI] = 1.18 [-0.17–2.53], Z = 1.71, P = 0.09) (**Fig 6**).

## Subgroup analyses

Subgroup analysis based on the continent showed that the overall AGE level was associated with GDM for studies conducted in Asia (SMD [95% CI] = 4.04 [2.48–5.59], Z = 5.09, P < 0.0001) and Europe (SMD [95% CI] = 1.51 [0.72–2.30], Z = 3.74, P = 0.0002) but not in those conducted in Latin America (SMD [95% CI] = -0.10 [-0.29–0.09], Z = 1.06, P = 0.29). Gestational period-based analysis revealed that the overall level of AGEs was significantly higher at the third trimester of pregnancy (SMD [95% CI] = 3.84 [2.31–5.37], Z = 4.92, P < 0.00001) but not at first and second trimesters of pregnancy. Regarding the type of samples collected, the blood/plasma AGE level was significantly higher among GDM (SMD [95% CI] = 2.86 [1.84–3.89], Z = 5.48, P < 0.00001) but not in AGE samples collected from skin and other sources (SMD [95% CI] = 0.36 [-0.25–0.97], Z = 1.15, P = 0.25) compared to control (**Table 3**).

Table 1. Study characteristics of diabetic and non-diabetic pregnant women included in the study.

| Authors | Year of publication | Country | Study Settings | Study design | BMI (Kg/m$^2$) | | Trimester of pregnancy | # Cases/ controls | Sample | Level of AGEs (AU/mL) | |
|---|---|---|---|---|---|---|---|---|---|---|---|
| | | | | | GDM | Controls | | | | GDM | Controls |
| | | | | | Mean ±SD | Mean ±SD | | | | Mean ±SD or Mean [95% CI] | Mean ±SD or Mean [95% CI] |
| Aziz et al [37] | 2015 | Iraq | Hospitals in Baghdad | Case-control | 28.80±0.27 | 27.77±0.40 | Third | 30/30 | Blood | **5.29±0.28** | **2.76±0.33** |
| Bartakova et al [30] | 2016 | Czek | Hospital of Brno | Case-control | 24.80 ± 1.76 | 21.15± 1.73 | Second | 182/36 | Blood | 10.18 [8.09–12.22] | 5.73 [5.29–8.66] |
| Boutzios et al [31] | 2013 | Greece | Aretaieion University Hospital, Athens | Cross-sectional | 30.3 ± 4.4 | 27.9 ± 3.1 | Second | 54/56 | Blood | 7.28 ± 2.4 | 5.68 ± 1.3 |
| Cosson et al [32] | 2019 | France | Jean-Verdier hospital, Paris | Case-control | NR | NR | Second | 48/70 | Skin | 1.99 ± 0.47 | 1.79 ± 0.32 |
| | | | | | | | First | 62/70 | | 2.11 ± 0.48 | |
| Davison et al [33] | 2011 | UK | Royal Liverpool University Hospital | Case-control | NR | NR | Third | 15/29 | Blood | 3.58±0.83 | 3.13±0.51 |
| de Ranitiz-Greven et al [34] | 2012 | Netherlands | University Medical Center Utrecht | Cross sectional | 27.6 ± 6.0 | 25.0 ± 6.3 | Second | 60/44 | Skin | 1.73 ± 0.33 | 1.81± 0.30 |
| Harsem et al [35] | 2008 | Norway | Ulleval University Hospital. | Case-control | 32.1 ± 4.33 | 28.5 ± 4.89 | Third | 34/38 | Blood | 1.72 (1.47–1.91) | 1.28 (1.13–1.42) |
| Krishnasamy et al [39] | 2019a | India | Tertiary referral centers in Tamil Nadu | Case-control | 25.72 ± 5.48 | 24.22 ± 4.67 | Third | 50/50 | Blood | 13.18 ± 8.74 | 2.68 ± 0.89 |
| Krishnasamy et al [40] | 2019b | India | Two tertiary care hospitals | Case-control | 26.36 ± 0.61 | 26.27 ± 0.25 | Third | 50/50 | Blood | 10.40 ± 0.98 | 4.71 ± 0.39 |
| Li et al [41] | 2019 | China | Binzhou City Center Hospital | Case-control | 23.17±3.16 | 22.86±2.66 | Second | 72/80 | Placenta | 54.27±18.28 | 32.18±12.12 |
| Li and Yang [42] | 2019 | China | Peking University First Hospital | Case-control | 22.71 ± 3.20 | 21.65 ± 2.84 | Second | 90/90 | Blood | 473.65 ± 105.32 | 324.36 ± 57.86 (ng/L)* |
| | | | | | | | Third | 90/90 | Blood | 533.47 ± 146.95 | 315.50 ± 77.79 (ng/L)* |
| Lobo et al [44] | 2017 | Brazil | NS | Case-control | 32.0 ± 2.22 | 25.4 ± 1.48 | Second | 225/217 | Blood | 2.42 ± 0.72 | 2.50 ± 0.86 |
| Pertyn´ska-Marczewska [36] | 2009 | Poland | Polish Mother's Memorial Hospital | Case-control | 24.25 ± 2.57 | 23.17 ± 4.43 | First | 14/14 | Blood | 9.5 ± 1.9 | 5.2 ± 1.3 |
| | | | | | NR | NR | Third | 14/14 | Blood | 9.7 ± 1.9 | 5.3 ± 0.7 |
| Guosheng et al [38] | 2009 | China | First Affiliated Hospital of Jinan University | Case-control | NR | NR | Second | 60/72 | Blood | 8.114±2.375 | 4.262±1.284 |
| | | | | | | | Third | 72/80 | Blood | 8.085±2.396 | 4.830±1.156 |
| Mai et al [43] | 2014 | China | Guangdong Women and Children Hospital | Case-control | 22.7±3.5 | 21.5±2.7 | NS | 190/80 | Blood | 403.0+ 208.6 | 321.8 +150.3 (ng/L) * |

(*Continued*)

**Table 1.** (Continued)

| Authors | Year of publication | Country | Study Settings | Study design | BMI (Kg/m²) | | Trimester of pregnancy | # Cases/ controls | Sample | Level of AGEs (AU/mL) | |
|---|---|---|---|---|---|---|---|---|---|---|---|
| | | | | | GDM | Controls | | | | GDM | Controls |
| | | | | | Mean ±SD | Mean ±SD | | | | Mean ±SD or Mean [95% CI] | Mean ±SD or Mean [95% CI] |
| Morales et al [45] | 2016 | Mexico | Regional general hospital | Cross-sectional | 33.99±5.32 | 30.09±4.02 | NS | 38/38 | Blood | NR | NR |

NS, not specified; NR, not recorded; SD, standardized difference

*measurements in units other than specified one. NB: Studies reported in 95% CI were converted to SD using STATA 15.0 for the sake of analysis.

## Meta-regression and sensitivity analysis

Strong evidence of high heterogeneity (Tau² = 1.95, I² = 97%, P< 0.0001) was demonstrated in the analysis of the relationship between the level of AGEs and GDM. For this, univariate meta-regression was run to identify potential covariates that likely affect the magnitude and direction of the overall SMD estimate. Nonetheless, meta-regression with the sample size (sum of cases and controls) (regression coefficient (Q) = -0.0092, P = 0.207) and year of publication (Q = 0.0035, P = 0.984) suggested that the covariates had no significant effect on the heterogeneity between studies. Likewise, the bivariate meta-regression analysis also showed that either of them did not significantly contribute to the heterogeneity (P = 0.65 and P = 0.20 for

**Table 2. Level of specific AGEs and related metabolic biomarkers implicated for diabetic complications.**

| Authors | Level of specific AGE and related metabolic biomarkers | | Type of marker |
|---|---|---|---|
| | GDM | Controls | |
| | Mean ±SD/Mean [95% CI] | Mean ±SD/Mean [95% CI] | |
| Aziz et al | 1.93±0.26 | 1.43±0.26 | HOMA-IR |
| | 6.51±0.33 | 5.07±0.08 | HbA1c (%) |
| Bartakova et al | 683.64 [560.00–805.30] | 507.54 [433.56–679.50] | CML (ng/ml) |
| Boutzios et al | 2.33 ±3.34 | 1.63 ±1.44 | HOMA-IR |
| | 5.4 ±0.47 | 5.35 ±0.37 | HbA1c (%) |
| Davison et al | 6.15±0.71 | 5.39±0.39 | HbA1c (%) |
| Harsem et al | 2.18 (2.10–2.66) | 2.49 (2.37–3.19) | CML (U/ml) |
| Krishnasamy et al | 217.8 ± 86.92 | 142.3 ± 38.21 | ICAM-1 (ng/ml) |
| | 15.7 ± 13.54 | 9.26 ± 5.38 | MGO (ng/ml) |
| Krishnasamy et al | 201.04 ± 7.85 | 174.1 ± 7.11 | ICAM-1 (ng/ml) |
| Li et al | 6.21±1.03 | 3.87±0.71 | MDA (ng/ml) |
| Li and Yang | 2.53 ± 1.92 | 1.84 ± 1.21 | HOMA-IR |
| Pertyn´ska-Marczewska | 825 (642–884 | 1231 (1040–1586) | SRAGE (pg/ml) |
| | 705 (555–885) | 985 (822–1221) | SRAGE (pg/ml) |
| Mai et al | 5.7 ±0.5 | 5.5 ± 0.3 | HbA1c (%) |
| | 1.9 ±1.2 | 1.5 ±0.9 | HOMA-IR |
| Morales et al | 80.47±63.3 | 79.4±36.76 | ICAM-1 (ng/ml) |
| | 427.58±72.7 | 420.58 ± 97.43 | VCAM-1 (ng/ml) |
| | 2.70 ±1.56 | 1.96 ±1.08 | HOMA-IR |

AGE, Advanced glycation end products; HOMA-IR, Homeostatic assessment for insulin resistance; ICAM-1, Intercellular adhesion molecule-1; VCAM-1, Vascular cell adhesion molecule-1; HbA1c, glycated hemoglobin; SRAGE, soluble receptor for AGEs; MDA, malondialdehyde; MGO, methylglyoxal; CML, Carboxymethyl-leucine.

| Study or Subgroup | GDM Mean | SD | Total | Controls Mean | SD | Total | Weight | Std. Mean Difference IV, Random, 95% CI | Std. Mean Difference IV, Random, 95% CI |
|---|---|---|---|---|---|---|---|---|---|
| Aziz et al 2015 | 5.29 | 0.28 | 30 | 2.76 | 0.33 | 30 | 5.7% | 8.16 [6.57, 9.75] | |
| Boutzios et al 2013 | 7.28 | 2.4 | 54 | 5.68 | 1.3 | 56 | 7.5% | 0.83 [0.44, 1.22] | |
| Cosson et al 2019 | 2.11 | 0.48 | 48 | 1.79 | 0.32 | 70 | 7.5% | 0.81 [0.43, 1.19] | |
| Cosson et al' 2019 | 1.99 | 0.47 | 48 | 1.79 | 0.32 | 70 | 7.5% | 0.51 [0.14, 0.89] | |
| Davison et al 2011 | 3.58 | 0.83 | 15 | 3.13 | 0.51 | 29 | 7.3% | 0.70 [0.05, 1.34] | |
| de Ranitiz-Green et al 2012 | 1.73 | 0.33 | 60 | 1.81 | 0.3 | 44 | 7.5% | -0.25 [-0.64, 0.14] | |
| Guosheng et al 2009 | 8.11 | 2.4 | 60 | 4.26 | 1.28 | 72 | 7.5% | 2.04 [1.62, 2.47] | |
| Guosheng et al' 2009 | 8.08 | 2.39 | 72 | 4.83 | 1.156 | 80 | 7.5% | 1.75 [1.38, 2.13] | |
| Harsem et al 2008 | 1.72 | 0.11 | 34 | 1.28 | 0.07 | 38 | 6.9% | 4.78 [3.85, 5.71] | |
| Krishnasamy et al 2019a | 13.18 | 8.74 | 50 | 2.68 | 0.89 | 50 | 7.5% | 1.68 [1.22, 2.14] | |
| Krishnasamy et al 2019b | 10.4 | 0.98 | 50 | 4.71 | 0.39 | 50 | 6.5% | 7.57 [6.43, 8.71] | |
| Lobo et al 2017 | 2.42 | 0.72 | 225 | 2.5 | 0.86 | 217 | 7.6% | -0.10 [-0.29, 0.09] | |
| Pertyn´ ska-Marczewska 2009 | 9.5 | 1.9 | 14 | 5.2 | 1.3 | 14 | 6.7% | 2.56 [1.53, 3.60] | |
| Pertyn´ ska-Marczewska' 2009 | 9.7 | 1.9 | 14 | 5.3 | 0.7 | 14 | 6.6% | 2.98 [1.86, 4.11] | |
| **Total (95% CI)** | | | 774 | | | 834 | 100.0% | 2.26 [1.50, 3.02] | |

Heterogeneity: Tau² = 1.95; Chi² = 505.41, df = 13 (P < 0.00001); I² = 97%
Test for overall effect: Z = 5.83 (P < 0.00001)

**Fig 2. Forest plot depicting the mean level (U/ml) of AGEs in GDM and pregnant controls.**

publication year and sample size, respectively). The bubble plots of SMD with sample size and publication year are presented in **Figs 7 and 8,** respectively. In the sensitivity analysis, three studies were removed 'turn by turn' and 'all at once' but no significant change was observed on the degree of heterogeneity and the pooled SMD remained significant in all analyses. In case of HOMA-IR and HbA1C sensitivity analysis, the "leave-one-out" sensitivity analysis indicated that excluding one outlier study abolished heterogeneity in HOMA-IR studies ($I^2$ = 0.0%, P = 0.81) (Fig 4) and showed a substantial but non-significant reduction in heterogeneity of studies reporting HbA1C ($I^2$ = 81%, P = 0.005) (Fig 5).

## Influence analysis and publication bias

In the influence analysis, no single study had excessive influence on the relationship between AGE and GDM (the pooled SMD estimate) (**Fig 9**). To confirm a small study effect, Egger's regression test accompanied with funnel plot asymmetry demonstrated that there was a sort of publication bias (Egger's Q = 10.11, P < 0.022) (**Fig 10**).

## Discussion

In this meta-analysis, we investigated the relationship between the overall level of AGEs and GDM. The pooled SMD estimate revealed that the level of AGEs was significantly higher in women with GDM than controls. Besides, the mean BMI of women with GDM showed a significant difference compared with controls. Subgroup analyses indicated that studies from Asia and Europe, outcome measures at the third trimester of pregnancy, and blood/plasma AGE samples showed a statistically significant mean difference between women with GDM and pregnant controls.

Notwithstanding the presence of several reports that demonstrate the relationship between AGEs, hyperglycemia, and oxidative stress, the underlying mechanism of causal relationship between body AGEs and GDM remains equivocal. It has become evident that AGEs increase

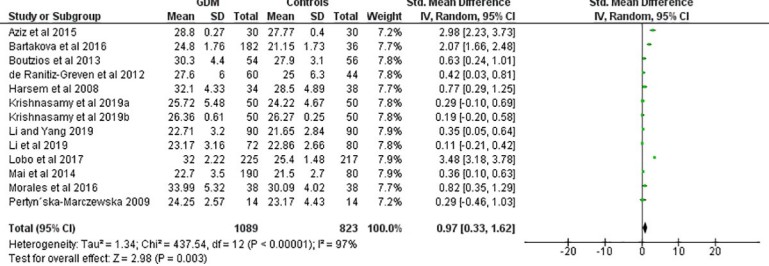

| Study or Subgroup | GDM Mean | SD | Total | Controls Mean | SD | Total | Weight | Std. Mean Difference IV, Random, 95% CI | Std. Mean Difference IV, Random, 95% CI |
|---|---|---|---|---|---|---|---|---|---|
| Aziz et al 2015 | 28.8 | 0.27 | 30 | 27.77 | 0.4 | 30 | 7.2% | 2.98 [2.23, 3.73] | |
| Bartakova et al 2016 | 24.8 | 1.76 | 182 | 21.15 | 1.73 | 36 | 7.7% | 2.07 [1.66, 2.48] | |
| Boutzios et al 2013 | 30.3 | 4.4 | 54 | 27.9 | 3.1 | 56 | 7.8% | 0.63 [0.24, 1.01] | |
| de Ranitiz-Green et al 2012 | 27.6 | 6 | 60 | 25 | 6.3 | 44 | 7.8% | 0.42 [0.03, 0.81] | |
| Harsem et al 2008 | 32.1 | 4.33 | 34 | 28.5 | 4.89 | 38 | 7.7% | 0.77 [0.29, 1.25] | |
| Krishnasamy et al 2019a | 25.72 | 5.48 | 50 | 24.22 | 4.67 | 50 | 7.8% | 0.29 [-0.10, 0.69] | |
| Krishnasamy et al 2019b | 26.36 | 0.61 | 50 | 26.27 | 0.25 | 50 | 7.8% | 0.19 [-0.20, 0.58] | |
| Li and Yang 2019 | 22.71 | 3.2 | 90 | 21.65 | 2.84 | 90 | 7.9% | 0.35 [0.05, 0.64] | |
| Li et al 2019 | 23.17 | 3.16 | 72 | 22.86 | 2.66 | 80 | 7.8% | 0.11 [-0.21, 0.42] | |
| Lobo et al 2017 | 32 | 2.22 | 225 | 25.4 | 1.48 | 217 | 7.9% | 3.48 [3.18, 3.78] | |
| Mai et al 2014 | 22.7 | 3.5 | 190 | 21.5 | 2.7 | 80 | 7.9% | 0.36 [0.10, 0.63] | |
| Morales et al 2016 | 33.99 | 5.32 | 38 | 30.09 | 4.02 | 38 | 7.7% | 0.82 [0.35, 1.29] | |
| Pertyn´ska-Marczewska 2009 | 24.25 | 2.57 | 14 | 23.17 | 4.43 | 14 | 7.2% | 0.29 [-0.46, 1.03] | |
| **Total (95% CI)** | | | 1089 | | | 823 | 100.0% | 0.97 [0.33, 1.62] | |

Heterogeneity: Tau² = 1.34; Chi² = 437.54, df = 12 (P < 0.00001); I² = 97%
Test for overall effect: Z = 2.98 (P = 0.003)

**Fig 3. Forest plot depicting the BMI of women with GDM and controls.**

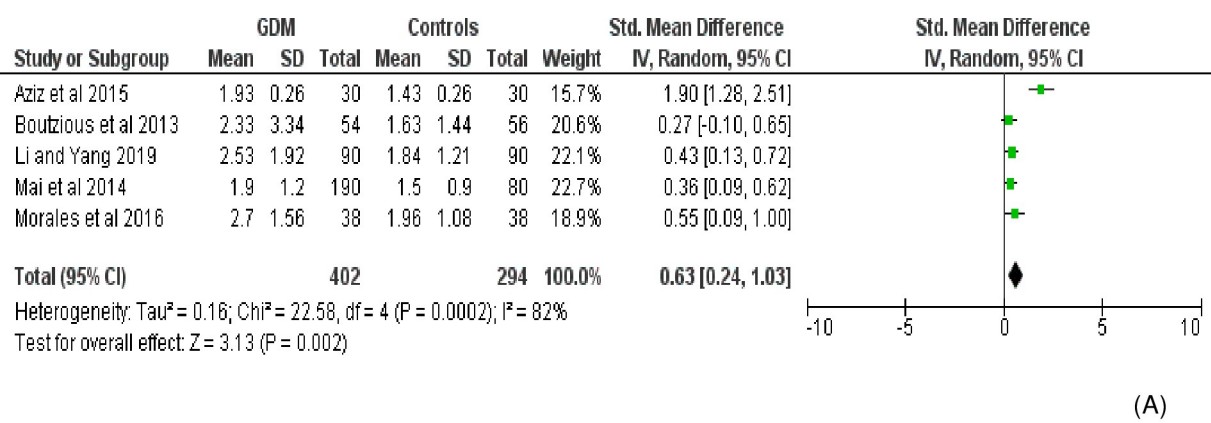

**Fig 4. Forest plot comparing the level of HOMA-IR among women with GDM and pregnant controls.** (A) without sensitivity analysis (B) with sensitivity analysis.

ROS formation and impair antioxidant systems. Thus, AGEs can partly contribute to chronic stress conditions in diabetes [46]. In turn, the formation of some AGEs is induced by states of oxidative stress which plays a more important role in the formation of AGEs in type 2 diabetes in which unhealth weight gain is commonly observed [47, 48]. Sustained load of these oxidants may surmount host defense mechanisms and lead to unopposed oxidative stress and chronic inflammation. Over time, these states can chronically impair insulin production and/or sensitivity and lead to diabetes. Moreover, hyperglycemia is a major driving force for AGE formation, especially when there is a pre-existing oxidative stress. It is becoming increasingly vivid that the contemporary diets are loaded with preformed AGEs, which may catalyze oxidative stress [11]. There are several research reports that link the high level of AGEs in the body with metabolic syndrome, type 2 diabetes, and cardiovascular diseases [49, 50], diabetic macrovascular diseases [51], greater cognitive decline in older adults [52], and new or worsening nephropathy [53].

Even though much is yet to be investigated, studies suggested that interaction of AGEs with RAGE alters downstream signaling pathways and results in gene expression, release of pro-inflammatory molecules and free radicals [7]. The AGE/RAGE axis may also play a pivotal role in the arterial calcification of diabetes through various mechanisms [54]. Hence, blockade of AGEs formation or interaction with RAGE and suppressing downstream signaling pathways have become viable targets in the treatment of diabetes and metabolic syndrome [55, 56]. A study revealed that AGE-mediated activation of early growth response protein 1 (EGR-1)

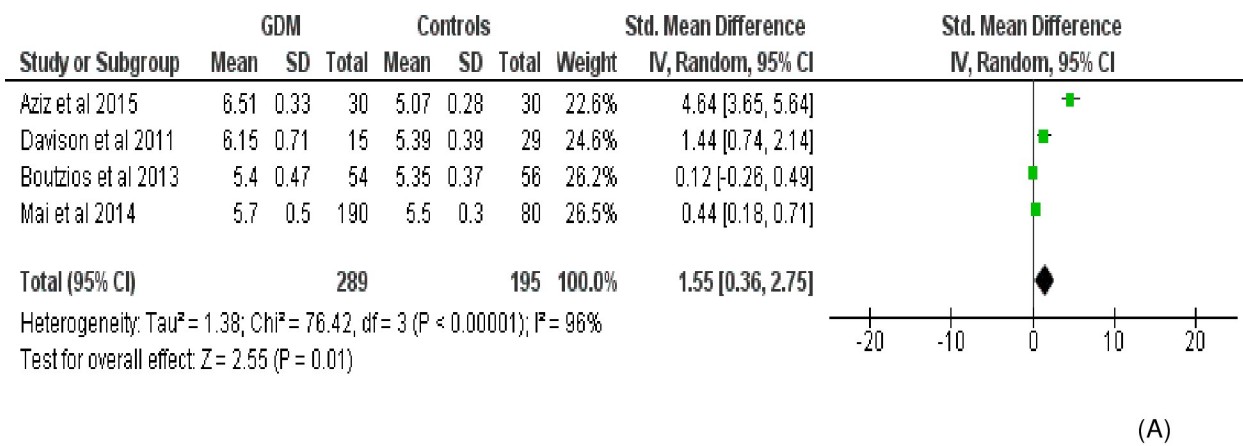

**Fig 5. Forest plot depicting the level of HbA1c among women with GDM and controls.** (A) without sensitivity analysis (B) with sensitivity analysis.

and its downstream factors via protein kinase C- βII (PKC-βII) and extracellular signal-regulated kinase 1/2 (ERK1/2) signaling pathway (AGEs/PKC-βII/ ERK1/2/EGR-1 pathway), is a novel mechanism of inducing vascular inflammation in GDM [57]. AGE-RAGE interaction activates its downstream signaling pathways, such as nuclear factor (NF)-kB and phosphoinositide 3-kinase (PI3K)/Akt, ultimately leading to diabetes and cancers [18]. There is increasing evidence that supports the role of RAGE in the pathogenesis of type 1 diabetes. Hence, blockade of RAGE, its ligands or signal transduction presents a viable target for the secondary prevention of diabetes [58].

The pooled SMD estimate from studies reporting HOMA-IR and HbA1c indicated that there is a significant difference between women with GDM and controls. In line with this, in HIT-T15 cell lines cultured with AGEs, a reduced expression and nuclear localization of pancreatic and duodenal homeobox-1 (PDX-1) gene, a decreased phosphorylation, and an increased acetylation of transcription factor (FoxO1) was observed. Consequently, AGEs

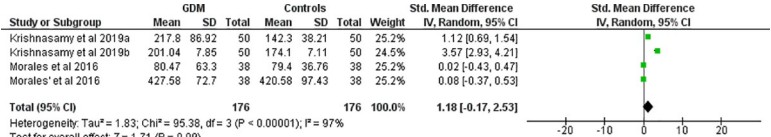

**Fig 6. Forest plot depicting the level of ICAM-1 and VCAM-1 (ng/ml) among women with GDM and controls.**

**Table 3. Subgroup analysis of outcome measures based on continent, trimester of pregnancy and sample type.**

| Variables and subgroups | | Effect size | | | Heterogeneity | | |
|---|---|---|---|---|---|---|---|
| | | SMD [95% CI] IV, random | Z-statistic | P-value | X² | I² (%) | p-value |
| Continent | Asia | 4.04 [2.48,5.59] | 5.09 | <0.0001 | 150.02 | 97 | <0.00001 |
| | Europe | 1.51 [0.72, 2.30] | 3.74 | 0.0002 | 126.58 | 94 | <0.00001 |
| | Latin America | -0.10 [-0.29, 0.09] | 1.06 | 0.29 | --- | ----- | ----- |
| Test for subgroup difference | | | | | 40.84 | 95.1 | <0.00001 |
| Trimester of pregnancy | First trimester | 1.62 [-0.10, 3.33] | 1.85 | 0.06 | 9.70 | 90 | 0.002 |
| | Second trimester | 0.60 [-0.14, 1.34] | 1.59 | 0.11 | 98.46 | 96 | <0.0001 |
| | Third trimester | 3.84 [2.31, 5.37] | 4.92 | <0.00001 | 202.09 | 97 | <0.00001 |
| Test for subgroup difference | | | | | 14.19 | 85.9 | 0.008 |
| Type of sample collected | Blood | 2.86 [1.84,3.89] | 5.48 | <0.00001 | 471.24 | 98 | <0.00001 |
| | Skin | 0.36 [-0.25,0.97] | 1.15 | 0.25 | 15.32 | 87 | 0.0005 |
| Test for subgroup difference | | | | | 16.95 | 94.1 | <0.0001 |

decrease insulin content through unbalancing the transcription factors and regulating insulin gene expression [59]. AGEs can also promote insulin resistance and hence trigger diabetes by depleting the antioxidant defenses such as AGE receptor-1 and a survival factor sirtuin-1 [8]. AGEs can undergo post-translational modification of insulin molecule and impair its function [60, 61]. Tan *et al.* reported that serum level of AGEs is linked with insulin resistance even in non-obese and non-diabetic subjects, reinforcing the notion that AGEs can be an independent determinant of HOMA-IR [62].

Regarding HbA1c and AGEs, a study indicated that HbA1c level was positively and significantly correlated with blood AGEs in obese Brazilian subjects [63]. Gestational weight gain and average third trimester HbA1c level (>5%) were found as risk factors for neonatal complications in mothers with GDM [64]. Likewise, measurement of HbA1c, at the end of pregnancy was associated with adverse pregnancy outcomes [65]. Piuri et al also observed a positive correlation between methylglyoxal (MGO) levels, the major precursor in the formation of AGEs, and HbA1c both at diagnosis and after 12 weeks of gestation. MGO was significantly

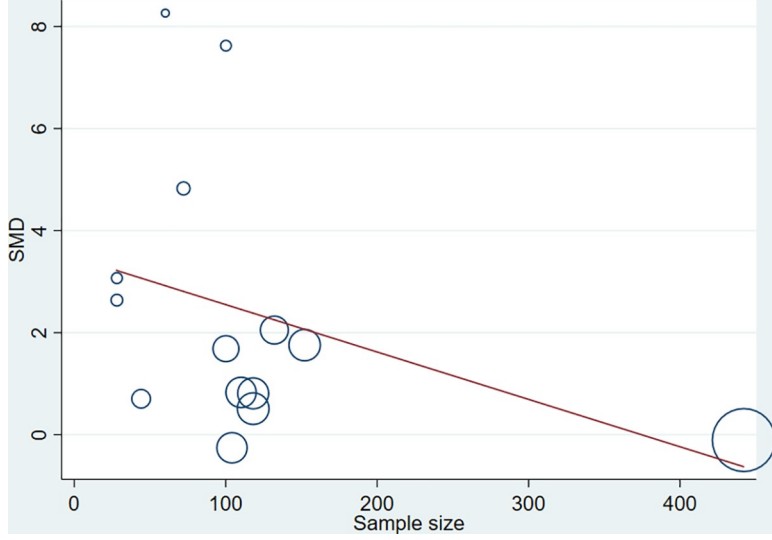

**Fig 7. Bubble plot depicting the univariate meta regression of SMD with overall sample size.**

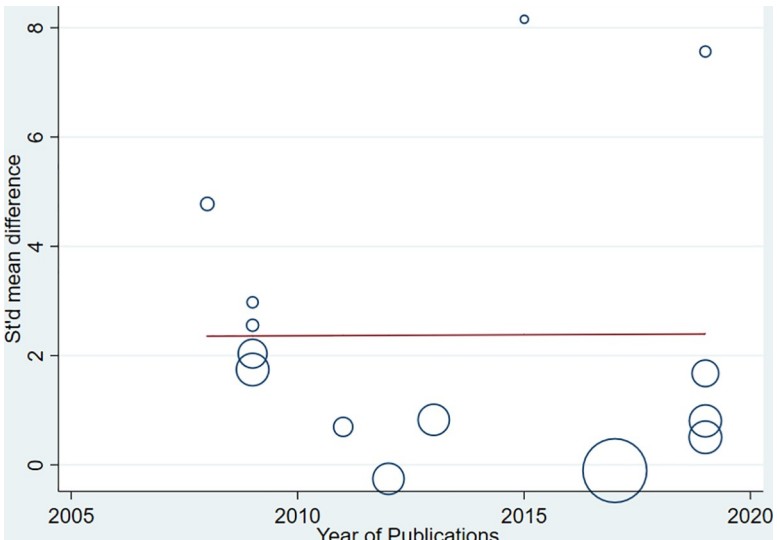

**Fig 8. Bubble plot depicting the univariate meta-regression of SMD with years of publication.**

correlated with the HOMA-IR index as well. MGO levels were also positively correlated with both the pregestational and gestational weight of women [19]. Hence, an HbA1c and HOMA-IR value determined during pregnancy can provide useful information and identify pregnancies that require fetal surveillance [66].

Elevated levels of endothelial cell adhesion molecules in GDM women indicate an imbalance in vascular function. Transient hyperglycemia may provoke a persistent modification to the memory cells and hence, women with GDM are more prone to develop future complications than controls. Likewise, an increased levels of ICAM-1, VCAM-1, and selectins in women with GDM are a reflection of endothelial dysfunction contemplating the future metabolic risks via metabolic memory effects [67–69]. Studies showed that circulating levels of AGEs were positively associated with severity of aortic calcification and diabetes-related

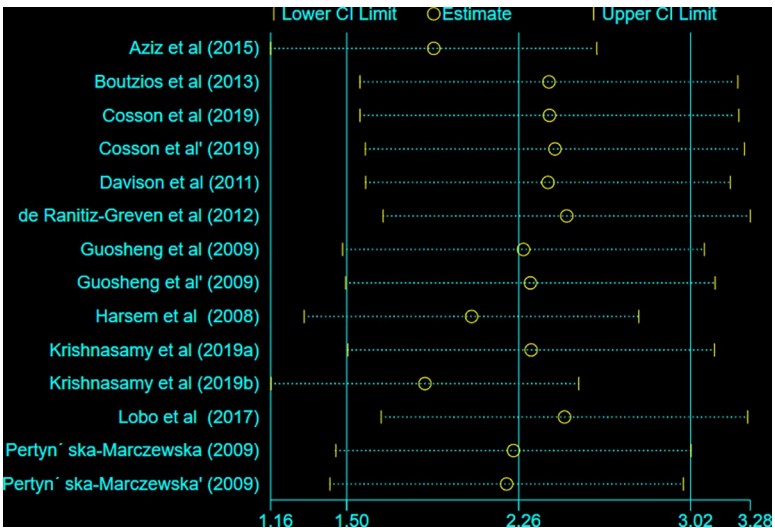

**Fig 9. Meta-influence plot showing the impact of every study on the overall SMD estimate.**

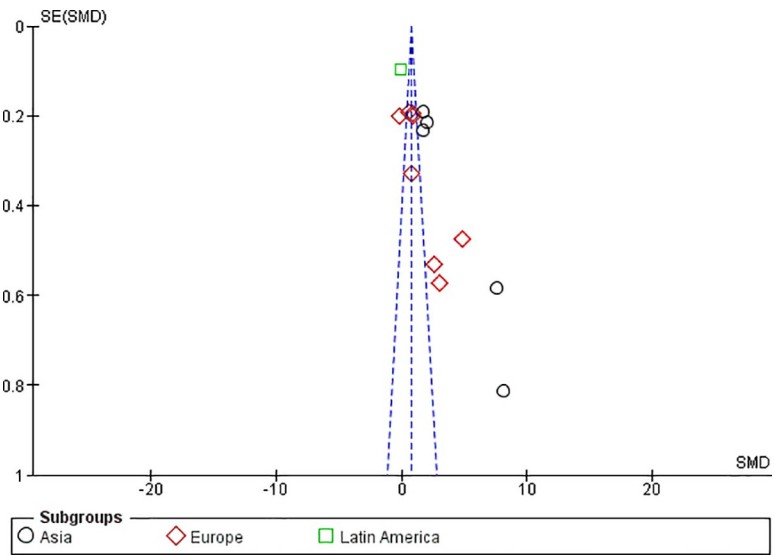

**Fig 10. Funnel plot showing the publication bias of studies.** Small studies with larger effect sizes are shown in the right lower side of the plot.

complications. Therefore, increased levels of AGEs can be considered as a biomarker for various vascular complications of diabetes [70, 71].

Observing the funnel plot asymmetry and Egger's statistical test, this study showed a sort of publication bias. It is evident that small studies with larger effect size are more likely to be published [72]. Due to this, few studies fall on the right lower side of the funnel plot indicating larger effect size with greater standard error. The SMD estimate by itself is more likely to distort funnel plot in publication bias assessment particularly when there are studies with small sample size leading to overestimation of the existence and extent of publication bias [73].

## Strength and limitations of the study

Having considered the variability in study characteristics and effect size measurements, we employed inverse variance method with random effects pooling model. Different instrumental scales and calibrations with diverse units of measurement for continuous data remain a challenge to get a comprehensive and aggregate result. For this, studies reporting uniform units of measurement were considered for this meta-analysis section. Hedge's g based SMD estimates were considered for pooling the outcome measures. As this systematic review and meta-analysis also reflects the methodological characteristics and outcome measures of individual studies, the relationship between AGE and GDM should be considered as 'non-causal'. This systematic review and meta-analysis should be seen in the context of such limitations.

## Conclusion

The findings indicated that there is a strong relationship between GDM and the level of AGEs in the body. Further analysis on the level of few related metabolic biomarkers revealed a significant difference between women with GDM and controls. Subgroup analyses also indicated that the third trimester of pregnancy and plasma samples were endowed with higher levels of AGEs among women with GDM than controls. It should be pointed out the cause-effect relationship between AGEs levels and GDM remains elusive. Hence, further well-designed studies should be conducted to find out the causal association between AGEs and GDM.

## Supporting information

**S1 Table. Completed PRISMA checklist.** The checklist highlights the important components addressed while conducting systematic review and meta-analysis from observational studies. (DOC)

**S2 Table. Data abstraction format with crude data.** The table presented the ways of data collection (study characteristics and outcome measures) in Microsoft excel format. (XLSX)

**S3 Table. Critical appraisal scores of included studies.** The table shows the risk of bias assessments of studies with regard to design, conduct and analysis. (DOCX)

## Acknowledgments

We extend our thanks to Haramaya University College of Health and Medical Sciences staffs who have technically supported us to realize this systematic review and meta-analysis.

## Author Contributions

**Conceptualization:** Mekonnen Sisay, Dumessa Edessa.

**Data curation:** Dumessa Edessa, Abraham Nigussie Mekuria, Alemu Gebrie.

**Formal analysis:** Mekonnen Sisay, Tilahun Ali, Abraham Nigussie Mekuria, Alemu Gebrie.

**Investigation:** Mekonnen Sisay, Tilahun Ali, Alemu Gebrie.

**Methodology:** Mekonnen Sisay, Dumessa Edessa, Tilahun Ali, Abraham Nigussie Mekuria, Alemu Gebrie.

**Software:** Mekonnen Sisay.

**Writing – original draft:** Mekonnen Sisay, Alemu Gebrie.

**Writing – review & editing:** Dumessa Edessa, Tilahun Ali, Abraham Nigussie Mekuria.

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
