## [Decision Letter · Decision Letter 0]

17 Sep 2020

PONE-D-20-19300

Advanced glycation end products and gestational diabetes mellitus and its complications: A systematic review and meta-analysis of observational studies

PLOS ONE

Dear Dr. Sisay,

Thank you for submitting your manuscript to PLOS ONE. After careful consideration, we feel that it has merit but does not fully meet PLOS ONE’s publication criteria as it currently stands. Therefore, we invite you to submit a revised version of the manuscript that addresses the points raised during the review process.

An expert in the field handled your manuscript, and we are very thankful for their time and efforts. Although interest was found in your review, some comments arose that require your attention. Please address ALL of the reviewer's comments in your revised manuscript.

We look forward to receiving your revised manuscript.

Kind regards,

Frank T. Spradley

Academic Editor

PLOS ONE

2. Please confirm that you have included all items recommended in the PRISMA checklist including the full electronic search strategy used to identify studies with all search terms and limits for at least one database.

Reviewers' comments:

Reviewer's Responses to Questions

**Comments to the Author**

1. Is the manuscript technically sound, and do the data support the conclusions?

Reviewer #1: Yes

2. Has the statistical analysis been performed appropriately and rigorously? 

Reviewer #1: Yes

3. Have the authors made all data underlying the findings in their manuscript fully available?

Reviewer #1: Yes

4. Is the manuscript presented in an intelligible fashion and written in standard English?

Reviewer #1: Yes

5. Review Comments to the Author

Reviewer #1: Well written and well designed manuscript. https://doi.org/10.1016/j.drudis.2020.07.003

Advanced glycation end products in diabetes, cancer and phytochemical therapy

Update ms with above reference.

Need to cite 2020 references

6. PLOS authors have the option to publish the peer review history of their article (what does this mean?). If published, this will include your full peer review and any attached files.

Reviewer #1: **Yes: **Ganji Purnachandra Nagaraju

---

## [Author Response · Author response to Decision Letter 0]

19 Sep 2020

Dear Editor,

Thank you for your constructive comments for updating this manuscript. 

Here are the changes we made in the body of the manuscript during revision. 

The reviewer recommended updating the manuscript with latest (2020) references. Accordingly, we incorporated six relevant references in the background and discussion sections including the one recommended by the reviewer. We have accepted the reviewer insights in amending the manuscripts to date. Kindly check the track changed manuscript, please. 

We have also undergone careful language (mainly punctuation, syntax and normalization issues) and technical editions throughout the manuscript to make it clearer and easier to comprehend

We have consistently formatted the ‘texts within figures’ to avoid ambiguity

We have strictly followed PLOS ONE formatting guidelines including 

o Correcting all figures using PACE (e.g. tif compatible format) with better resolution

o Rearranging the position of tables next to the paragraphs that they are first cited

o Formatting the title page, heading and subheadings of main text as per the guideline 

o Formatting the references consistently as per the PLOS ONE style (e.g. Vancouver with six authors followed by et al….)

The PRISMA flow diagram was followed as per the standard 

The full search strategy for legitimate databases such as PubMed, MEDLINE, and EMBASE was clearly and concisely presented in the methodology and reported in the PRISMA checklist. 

PRISMA Checklist was updated for the revised manuscript and all the contents of the checklist were adequately addressed in this systematic review and meta-analysis

We have further elaborated the eligibility criteria 

The final formatting including rearrangement of tables and figures was made in the clean version only (labeled as manuscript)

We have amended the title in the manuscript and online platform to be identical 

Regards, 

Authors

---

## [Decision Letter · Decision Letter 1]

25 Sep 2020

The relationship between advanced glycation end products and gestational diabetes : a systematic review and meta-analysis

PONE-D-20-19300R1

Dear Dr. Sisay,

We’re pleased to inform you that your manuscript has been judged scientifically suitable for publication and will be formally accepted for publication once it meets all outstanding technical requirements.

Kind regards,

Frank T. Spradley

Academic Editor

PLOS ONE

Reviewers' comments:

Reviewer's Responses to Questions

**Comments to the Author**

1. If the authors have adequately addressed your comments raised in a previous round of review and you feel that this manuscript is now acceptable for publication, you may indicate that here to bypass the “Comments to the Author” section, enter your conflict of interest statement in the “Confidential to Editor” section, and submit your "Accept" recommendation.

Reviewer #1: (No Response)

2. Is the manuscript technically sound, and do the data support the conclusions?

Reviewer #1: (No Response)

3. Has the statistical analysis been performed appropriately and rigorously? 

Reviewer #1: (No Response)

4. Have the authors made all data underlying the findings in their manuscript fully available?

Reviewer #1: (No Response)

5. Is the manuscript presented in an intelligible fashion and written in standard English?

Reviewer #1: (No Response)

6. Review Comments to the Author

Reviewer #1: The relationship between advanced glycation end products and gestational diabetes : a systematic review and meta-analysis. Revised manuscript acceptable

7. PLOS authors have the option to publish the peer review history of their article (what does this mean?). If published, this will include your full peer review and any attached files.

Reviewer #1: No

---

## [Editor Report · Acceptance letter]

28 Sep 2020

PONE-D-20-19300R1 

The relationship between advanced glycation end products and gestational diabetes : a systematic review and meta-analysis 

Dear Dr. Sisay:

I'm pleased to inform you that your manuscript has been deemed suitable for publication in PLOS ONE. Congratulations! Your manuscript is now with our production department. 

Kind regards, 

on behalf of

Dr. Frank T. Spradley 

Academic Editor

PLOS ONE